# Melatonin agonist tasimelteon (HETLIOZ®) improves sleep in patients with primary insomnia: A multicenter, randomized, double-blind, placebo-controlled trial

**Naoise C. Synnott**, **Christos M. Polymeropoulos**\*, **Changfu Xiao, Gunther Birznieks, Mihael H. Polymeropoulos**

Vanda Pharmaceuticals Inc., Washington, United States of America

\* christos.polymeropoulos@vandapharma.com

## Abstract

### Background

Tasimelteon is a dual melatonin 1 and melatonin 2 receptor agonist. Tasimelteon is the first and only approved medicine to treat a circadian rhythm disorder. In this Phase III trial, the efficacy and safety of tasimelteon was studied in primary insomnia characterized by difficulty falling asleep.

### Trial design

A randomized, double-blind, placebo-controlled, multi-center study investigated 20 mg or 50 mg tasimelteon vs placebo in 322 patients with primary insomnia over a 5-week double-blind treatment interval using polysomnography(PSG) measures of sleep.

### Methods

Patients underwent PSGs on Nights 1, 8, 22 and 29. Entry criteria emphasized enrollment of primary insomnia patients with a confirmed difficulty falling asleep. Subjective sleep latency was ≥ 45 minutes based on sleep history and sleep diary and, patients had a mean latency to persistent sleep (LPS) of ≥30 minutes on two consecutive screening PSG nights with no night having an LPS less than 20 minutes.

### Findings

On the primary end point, the mean improvement in LPS from baseline to the average of Nights 1 and 8 was 44.9 minutes (20 mg) and 46.3 minutes (50 mg) versus 28.2 minutes (placebo) (p < 0.001). Improvements in LPS persisted through the follow-up time points (Nights 22 and 29, p < 0.01). Tasimelteon use was not associated with cognitive or mood changes, and neither rebound nor withdrawal effects were observed after discontinuation.

**Data availability statement:** Human research participant data beyond what is provided in this paper is ethically and legally restricted by Vanda Pharmaceuticals Inc., as the data were collected and used with the consent of individuals who participated in the clinical trial. Data access queries can be directed to Data Access Committee secretary, Vuk Koprivica, Ph.D. (Vuk.Koprivica@vandapharma.com).

**Funding:** This work was sponsored by Vanda Pharmaceuticals. The funders designed and conducted the study, analyzed the data, and prepared the manuscript for publication.

**Competing interests:** All authors are employees of Vanda Pharmaceuticals.

## Conclusion

Tasimelteon improved sleep from the first night of treatment, and the effect continued for the duration of the study. Tasimelteon was well-tolerated with no adverse next-day residual effects observed. The results of the study strongly suggest that tasimelteon may be an effective therapeutic tool in the treatment of individuals with chronic sleep onset insomnia.

## Introduction

Disorders of sleep and wakefulness, which as a group are reported to chronically affect about 50 to 70 million Americans, comprise many distinct conditions [1]. However, insomnia is the most common clinical sleep disorder complaint. Insomnia is characterized as the difficulty in initiating or maintaining sleep or experiencing non-restorative or poor-quality sleep, and it is associated with next day consequences [2]. Insomnia can occur at different times of the night such as early in the night where there is an observed difficulty falling asleep ("sleep onset insomnia"), in the middle of the night where predominantly frequent nighttime awakenings are experienced ("middle insomnia") or late in the night where undesired early morning awakening is experienced ("late insomnia"). In a telephone interview study of more than 25,000 adults, more than 10 percent of respondents were "quite or completely dissatisfied with sleep latency (at least 30 min) or reporting long sleep latency at sleep-onset as a major sleep problem" at least three nights per week [3]. Even though the exact prevalence of insomnia is hard to determine, it has been reported that insomnia disorder affects approximately 10–17% of the American population [4,5].

Although the molecular underpinnings of sleep-onset insomnia remain unclear, many insomnia patients with sleep onset insomnia have a misalignment of their circadian rhythms with respect to their desired bedtime. In fact, in a study of 77 patients with primary insomnia who had an average sleep latency of 44.9 minutes as compared to 8.2 minutes in 21 control individuals, 22 percent of the primary insomnia patients attempted to sleep either before or within an hour of their dim light melatonin onset (DLMO) as compared to 6 percent of controls [6]. Moreover, DLMO occurred more than an hour later in the insomnia patients as compared to controls and therefore this population was trying to sleep at a significantly earlier circadian phase as compared to controls. We hypothesize that such patients could benefit from a therapeutic intervention using a melatonin-receptor agonist that could exhibit both a circadian regulatory effect as well as a circadian phase-dependent sleep-promoting effect, especially in patients who were attempting to fall asleep in the wake-maintenance zone [7–9], as the circadian regulatory effect would work to rapidly phase shift the patient's endogenous circadian pacemaker driving the sleep-wake propensity rhythm to be aligned with desired bedtime [10]. While sedative insomnia medications have been shown to promote sleep, these agents do not regulate circadian rhythms [10,11]. Moreover, unlike melatonin-receptor agonists, sedative hypnotics do not reduce the firing of the suprachiasmatic nucleus (SCN) neurons that

promote wakefulness during the wake-maintenance zone [12,13]. Therefore, the use of traditional hypnotics in patients with early insomnia would only facilitate sleep by promoting sedation or reduction of wakefulness, rather than quieting the output of the SCN.

Tasimelteon belongs to the class of melatonin receptor agonists. Because of its direct association with sleep and its involvement in the control of the circadian rhythm, pharmacological regulation of the melatonin receptor system is a candidate for the treatment of sleep disorders like insomnia, in particular insomnia characterized by difficulty falling asleep. Tasimelteon is a dual MT1/MT2 melatonin agonist that at least in part exerts it s sleep promoting effect as a circadian regulator [14]. Evidence from clinical development of tasimelteon suggests it may work through both chronobiotic effects by regulating the circadian rhythm and also through soporific effects by promoting sleepiness [15–17]. Thus, we hypothesized that tasimelteon may be an effective treatment for sleep onset insomnia.

In the United States (US), HETLIOZ® (tasimelteon capsule) is approved for the treatment of the Circadian Rhythm Sleep-Wake Disorder (CRSWD) Non-24-Hour Sleep-Wake Disorder (Non-24) in adults [18] and the treatment nighttime sleep disturbances in Smith-Magenis Syndrome (SMS) in patients 16 years of age and older [19]. HETLIOZ LQ® (tasimelteon oral suspension) is approved for the treatment of the nighttime sleep disturbances in SMS in pediatric patients 3 years to 15 years of age. Tasimelteon is the first and only approved medicine to treat Non-24 or sleep disturbances in SMS.

Here we present the results of a multicenter, randomized, double-blind, placebo-controlled Phase III trial in which the efficacy and safety of a single oral daily dose of tasimelteon in the treatment of primary insomnia patients was established.

## Materials and methods

The safety and effectiveness of tasimelteon (HETLIOZ®, Vanda Pharmaceuticals, Inc.) to treat chronic Insomnia Disorder was assessed in a placebo-controlled, double-masked, multicenter Phase III trial between 18 October 2007 and 13 March 2008. Patients were administered one of three treatment options: 20 mg tasimelteon, 50 mg tasimelteon, or placebo. The institutional review board at Copernicus Group (Cary, North Carolina, USA) reviewed and approved the study protocol (QUI1-07-404) and provided ethical oversight for the study. The trial was registered on ClinicalTrials.gov (Identifier: NCT00548340) prior to the enrollment of the first patient (first posted to ClinicalTrials.gov on October 23, 2007), with the primary outcome pre-specified as the average change from baseline in Latency to Persistent Sleep (LPS) to average of Treatment on Night 1 and Treatment on Night 8. The study was initiated at 39 US sites. All participants provided written informed consent prior to enrollment, and all methods were carried out in accordance with relevant guidelines and regulations.

### Study design

The study was divided into 2 phases: the pre-randomization phase and the randomization phase (Fig 1B). The pre-randomization phase consisted of the screening visit (12–21 days prior to start of evaluation phase) and a one week single-blind placebo lead-in that included two consecutive nights of PSG assessments. Upon completion of the screening visit, potential patients were given a sleep diary and a post-sleep questionnaire (PSQ) to take home with them. Patients were asked to answer the questions in the sleep diary every morning within one hour of waking up. The randomization phase consisted of a 5-week double-blind evaluation interval and a 1-night single-blind placebo wash-out night. During the double-blind evaluation interval, patients underwent four overnight visits [Nights 1, 8, 22 and 29 (± 2 days)] in which PSG was assessed. The scheduled bedtime in the sleep clinic was calculated as 8 hours prior to the patients habitual wake times during his/her normal work week. Patients returned to the clinic on Night 36 for an additional night of PSG assessments with single-blind placebo treatment.

### Patients

Men and women aged 18–64 years, who reported subjective sleep latency of ≥ 45 minutes at least 3 nights/week and subjective total sleep time ≤ 6.5 hours of sleep at least 3 nights/week, were eligible for screening. Patients who met

**(a)**

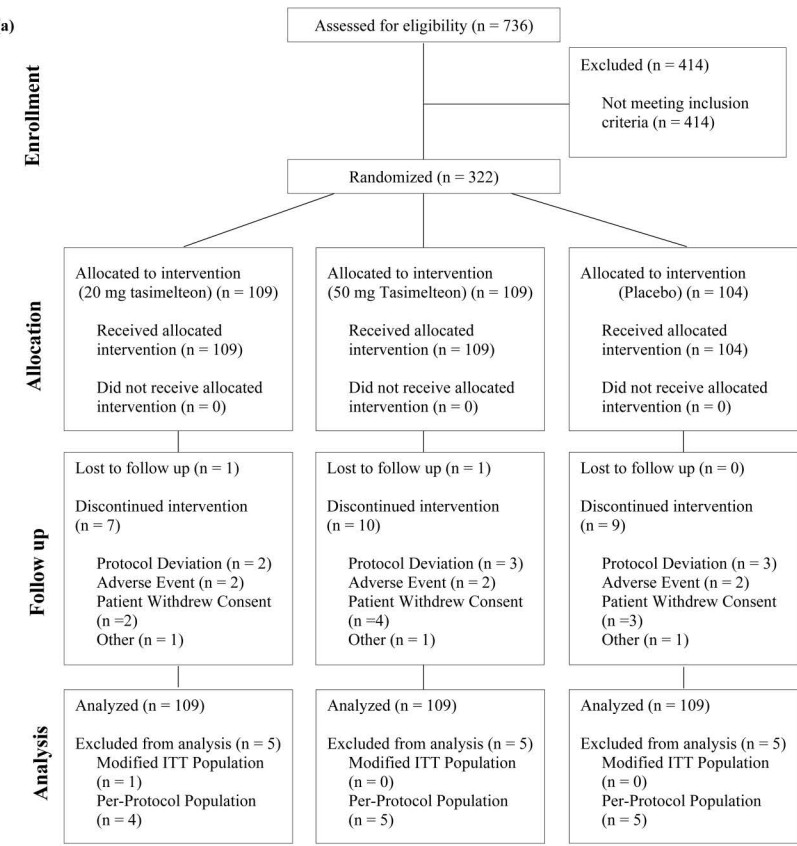

**(b)**

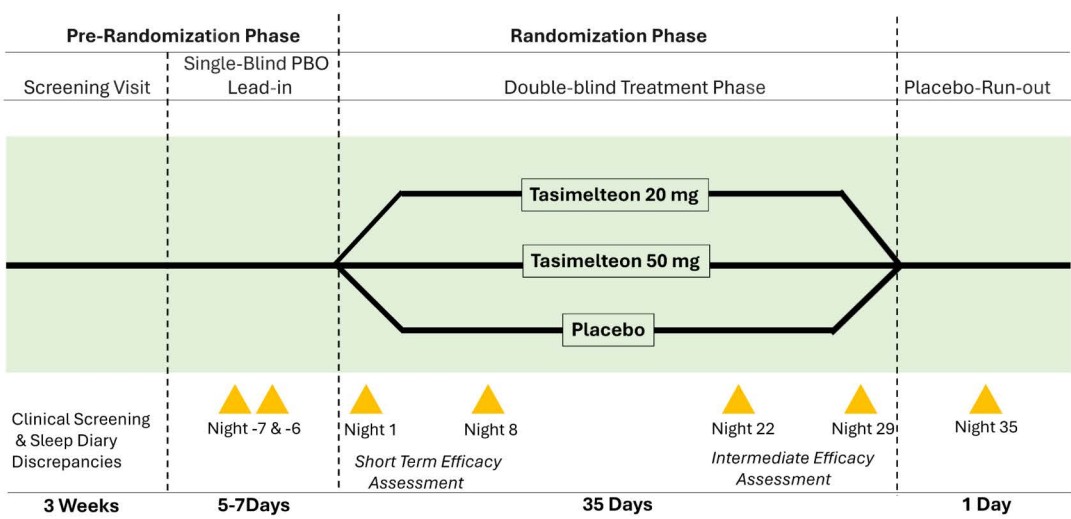

Study medication was taken 30 min before lights off/planned bedtime; In clinic scheduled bedtime was calculated as 8 hours prior to the patient's typical wake times during the normal work week; PBO= placebo.

▲ = In clinic polysomnography& pre-sleep questionnaire

**Fig 1. Patient Flow Diagram and Schema of Study Design.** (a) CONSORT patient flow diagram. (b) The study consisted of a 4-week pre-randomization phase followed by a randomization phase where insomnia patients were dosed with tasimelteon 20 mg, tasimelteon 50 mg or placebo 30 min prior to bedtime. The pre-randomization phase consisted of the screening visit and a 1 week single-blind placebo lead-in that included

2 consecutive nights of polysomnography (PSG) assessments. The randomization phase consisted of a 5-week double-blind treatment phase and a 1-night placebo run-out. In clinical PSG and a pre-sleep questionnaire were performed on nights 1, 8, 22, 29 and 36 of this phase. PBO = placebo.

diagnostic criteria for primary insomnia (as defined in DSM-IV) and had mean latency to persistent sleep (LPS) of ≥ 30 minutes on 2 consecutive placebo lead-in polysomnograms (PSG), were eligible for randomization. Persistent sleep was defined as the point at which 10 minutes (20 epochs) of uninterrupted sleep had begun.

Patients were also required to have a habitual bedtime between 9:00 PM and 1:00 AM for at least one month prior to screening.

Patients with a history of smoking, drug or alcohol abuse, psychiatric disorders, chronic obstructive pulmonary disease, other sleep disorders or irregular sleep/work schedules, cardiovascular issues, liver problems, intolerance to melatonin or other experimental drugs, or any other sound reason as determined by investigator, were excluded from the study.

Patients underwent physical examinations upon initial screening, at randomization and end-of study, including vital signs, hematology, blood chemistry and urinalysis, endocrine function and electrocardiogram, and were free of any major clinically significant medical, sleep or psychiatric condition unless currently controlled and stable, and were not taking sedative or stimulant central nervous system-active medications. The Benzodiazepine Withdrawal Symptom Questionnaire (BWSQ) was completed at the end of the double-blind and placebo-washout intervals.

### Randomization and masking

Patients were randomized centrally via the automated interactive voice response system (IVRS), a validated system that automates the random assignment of treatment groups to randomization numbers. The computer-generated randomization schedule was prepared and maintained by Clinphone Inc. prior to the start of the study. Randomization was balanced at the site level and utilized a randomization block size of 6 consisting of 2 randomization numbers from each of the 3 treatment groups: the placebo treatment group, the 20-mg VEC-162 treatment group, and the 50-mg VEC-162 treatment group. The randomization code was not provided to the Investigators or to Vanda until after the clinical database was locked.

### Procedure

**Single-blind placebo lead-in interval.** This interval consisted of 2 consecutive nights at the sleep clinic followed by 5 days of outpatient treatment. At the beginning of the placebo lead in, the sleep diary data from the seven consecutive days immediately preceding Visit 2 was reviewed to confirm eligibility, i.e., completed for at least 4/7 days. On the first night instructions, and a practice session, were given to the patients on how to complete the patient assessments i.e., Digit Symbol Substitution Test (DSST) and Visual Analogue Scale (VAS). In addition, the patients took an alcohol breathalyzer test and completed the pre-sleep questionnaire (Pre-SQ) prior to receiving single-blind placebo capsules ~30 min prior to their scheduled bedtime each inpatient night.

The first PSG was a diagnostic PSG (dPSG) which was also used to exclude patients with any other sleep disorder (e.g., sleep apnea). Patients were awakened at the end of the 8-hr sleep episode and were asked to complete patient assessments of sleep after 1 hr. Upon completion of the second PSG recording, patients were given a bottle containing single-blind placebo capsules and were instructed to take the study medication ~30 min before their planned bedtime every night until they returned to the clinic. The average of the two consecutive PSG results was used as baseline.

**Double-blind evaluation interval.** On each night of the double-blind evaluation interval at the sleep clinic, patients arrived at the sleep clinic 2.5 hours prior to their scheduled bedtime for pre-dose safety assessments. Patients who continued to meet the eligibility criteria on night 1 were then randomized. Patients were required to complete the Pre-SQ

prior to study drug administration and received double-blind study medication 30 min prior to scheduled bedtime and PSG was recorded for 8 hours. Patients were awakened at the end of the 8-hour sleep episode and were asked to complete the PSQ, VAS, and DSST, in this order, 1 hour after awakening. Prior to discharge from the sleep lab, patients were dispensed study medication under double-blind conditions and instructed to take 1 capsule every night 30 min prior to their planned bedtime. Furthermore, the patients were notified that for their safety they should only take their study medication if they have at least a 6-hour opportunity to sleep. Patients were also given a post-sleep questionnaire (PSQ) to take home with them. This outpatient PSQ was to be completed by the patient on the morning of the next scheduled study visit. At each of these visits, the patients were dispensed enough medication to last until the next scheduled visit. On Day 30, patients were asked to complete the BWSQ prior to patient assessments of sleep.

**Single-blind placebo wash-out night.** Patients were requested to return to the clinic on Night 36 for an additional night of sleep assessment following the procedure used for the evaluation interval. End-of-study (EOS) assessments were performed on the morning of Day 37. EOS assessments included safety evaluations and the completion of the BWSQ followed by the patient assessments 1 hour after awakening.

**Outcomes.** The pre-defined primary efficacy endpoint was the change from baseline of latency to persistent sleep (LPS) as measured by polysomnography for the average of Nights 1 and 8. Secondary endpoints included the assessment of 20 mg and 50 mg tasimelteon on the average wake after sleep onset (WASO), wake after sleep onset through hour 6 (WASOTO6), total sleep time (TST), and sleep efficiency (SE) for the average of Nights 1 and 8, also measured by polysomnography. The maintenance of the effect of oral doses of 20 mg and 50 mg tasimelteon on LPS, WASO, TST and SE was also measured by polysomnography on Nights 22 and 29.

Subjective sleep latency (sSL), subjective total sleep time (sTST), subjective sleep quality (sSQ), and subjective depth of sleep were measured by patient assessment on Days 2, 9, 23, and 30 by an in-patient PSQ. The visual analog mood scale (VAS) is a self-rated scale designed to assess feelings, affect, and mood, and their changes. Cognitive performance (functionality) was measured by the digital symbol substitution test (DSST), which is defined as the number of items correctly substituted within 90 seconds. Patient assessments were used to assess the effects of single and multiple oral doses of 20 mg and 50 mg tasimelteon.

Subjective daytime function and subjective daytime alertness were measured by Pre-SQ administered prior to receiving study drug on Nights 8, 22, 29, and 36 reflecting the subject's assessment of daytime function and alertness throughout the day. Potential for withdrawal symptoms (BWSQ), residual effects (DSST and VAS), and rebound effects (PSG) of these two doses were also investigated.

## Statistical analysis

**Sample size and power.** A t-test detecting a 15-minute difference in average sleep latency, with a standard deviation of 30 minutes, between tasimelteon treated patients and placebo patients was determined to need a sample size of 86 patients per treatment arm to have 90% power to detect a difference at a two-sided alpha of 0.05. Considering a 1:1:1 randomization between tasimelteon 20 mg, 50 mg and placebo, and a 20% drop-out rate, a total of 324 patients (108 per arm) was planned. Data was analyzed by Quintiles Canada, Inc. Biostatistical staff using SAS V.9.1.3.

Primary endpoint analysis was performed on the Modified Intent-To-Treat (ITT) population (n = 322), which included all randomized patients who received at least one dose of study medication and had post-baseline PSG data. Patient demographic and other baseline characteristics were calculated for all patients randomized (ITT). Safety analysis was performed on all patients who received study drug. Trial success was based on rejection of the null hypothesis associated difference in the mean change from baseline in the 2-night average (Nights 1 and 8). For the primary efficacy analyses, the LPS value was considered as a continuous variable. Estimates and tests for primary and secondary continuous efficacy variables were carried out by an Mixed-Model Repeated Measures (MMRM) model that includes 4 time points (Nights 1, 8, 22 and 29), three treatment groups (VEC 20 mg, VEC 50 mg and placebo), the effect for (pooled) site and the

baseline-by-time value as a covariate. The treatment by-baseline interaction term to check the consistency of treatment differences across various baseline score levels and treatment-by-(pooled) site to check the consistency of treatment differences across various (pooled) sites was tested using a 0.10 statistical significance level. All statistical tests were two-tailed. For the primary efficacy analysis, the type I error was controlled via Fisher's protected least significant difference (LSD). Using this procedure, an omnibus test of the overall treatment effect was conducted and pairwise comparisons were examined only if the overall treatment was significant at the 0.05 alpha level. Pairwise comparisons were also examined at the 0.05 significance level. Both the omnibus test and the pairwise comparisons will be conducted using the MMRM model described above.

ANCOVA was used to analyze subjective sleep assessments on LPS, WASO, TST and SE, as well as the potential for rebound effect of tasimelteon. For the ANCOVA model that includes main effects for treatment, (pooled) site and the baseline assessment as a covariate, the interaction between treatment and baseline term will be tested at the 0.10 alpha level to check the consistency of treatment differences across various baseline score levels.

## Results

A total of 322 patients with primary insomnia enrolled, were randomized into the study, and received double blinded medication. The Full Analysis (or modified ITT) population consisted of the 321 patients with insomnia who received at at least one dose of study medication and who had a post-baseline PSG evaluation. 294 patients (91.3%) completed the study (Fig 1A). A total of 28 patients (8.7%) withdrew from the study: 8 patients (7.3%) in the tasimelteon 20 mg treatment group; 11 patients (10.1%) in the tasimelteon 50 mg treatment group; and 9 patients (8.7%) in the placebo group. Among the patients who withdrew from the study, the most common reasons for discontinuation were withdrawal of consent (32.1%), protocol deviations (28.6%), and AEs (21.4%). Three patients discontinued the study for other reasons. No significant treatment group differences were noted for the individual reasons for discontinuation.

Baseline demographic characteristics were similar in the three treatment groups (Table 1). The mean age was 41.8 years with 125 males and 197 females. The majority of the patients were white (70.2%) and of non-Hispanic or Latino ethnicity (82.3%).

Table 1. Summary of Patient Demographic Information at Screening Visit.

| Demographic | | Placebo N = 104 | Tasimelteon 20 mg N = 109 | Tasimelteon 50 mg N = 109 | Total N = 322 |
|---|---|---|---|---|---|
| **Age (years)** | Mean (SD) | 41.7 (11.8) | 42.9 (10.6) | 40.7 (10.4) | 41.8 (10.9) |
| | Min, Max | 18, 62 | 20, 64 | 20, 62 | 18, 64 |
| **Sex, n (%)** | Male | 36 (34.6) | 43 (39.4) | 46 (42.2) | 125 (38.8) |
| | Female | 68 (65.4) | 66 (60.6) | 63 (57.8) | 197 (61.2) |
| **Race, n (%)** | Black or African American | 32 (30.8) | 27 (24.8) | 32 (29.4) | 91 (28.3) |
| | White | 71 (68.3) | 78 (71.6) | 77 (70.6) | 226 (70.2) |
| | Other | 1 (1.0) | 4 (3.6) | 0 | 5 (1.5) |
| **Ethnicity, n (%)** | Hispanic or Latino | 16 (15.4) | 22 (20.2) | 19 (17.4) | 57 (17.7) |
| | Non-Hispanic or Latino | 88 (84.6) | 87 (79.8) | 90 (82.6) | 265 (82.3) |
| **BMI (kg/m²)** | Mean (SD) | 26.0 (3.6) | 26.4 (3.6) | 26.8 (3.4) | 26.4 (3.5) |
| | Median | 26.70 (18.5, 33.0) | 26.60 (19.1, 32.7) | 25.45 (19.4, 32.9) | 26.50 (18.5, 33.01) |

Abbreviations: N = number; SD = standard deviation.

Note: Percentages are based on the total number of randomized patients with available data within each treatment group.

## Reduction in latency to persistent sleep

The primary endpoint of the trial was the effects of 20 and 50 mg tasimelteon on the average latency to persistent sleep (LPS) as measured by polysomnography on Nights 1 and 8.

Statistically significant improvements in LPS for the average of Nights 1 and 8 were observed for patients with insomnia receiving either 20 mg or 50 mg tasimelteon when compared with insomnia patients receiving placebo (p < 0.001 for both comparisons). The mean change (SEM) from baseline to the average of Nights 1 and 8 was −44.9 (2.9) min with 20 mg tasimelteon, −46.3 (2.9) min with 50 mg tasimelteon, and −28.2 (3.02) min with placebo (Table 2).

Statistically significant improvements in LPS persisted for the average of Nights 22 and 29 for patients receiving either 20 mg (p < 0.001) or 50 mg tasimelteon (p = 0.013). The estimate mean change (SEM) from baseline to the average of Nights 22 and 29 was 49.4 (3.3) minutes in the 20 mg tasimelteon group, 45.1 (3.3) minutes in the 50 mg tasimelteon treatment group, and 33.9 (3.3) minutes in the placebo group. Sensitivity analyses including an ANCOVA model evaluation of observed cases (OC) and Last Observation Carried Forward (LOCF) data and a randomization test (1000 replications) using the MMRM analysis model supported the conclusions of the original MMRM analyses for these parameters.

For individual night assessments, statistically significant improvement in LPS was noted for patients receiving 20 mg tasimelteon on all nights (Fig 2). For the subjective assessment of sleep latency, assessed the morning after in-clinic PSG recording by the PSQ, statistically significant improvements were observed on Day 2 for patients receiving either 20 mg tasimelteon (p < 0.001) or 50 mg tasimelteon (p = 0.024) when compared with subjects receiving placebo (Table 3).

## Improvements in sleep maintenance, duration and quality

Numerically greater improvements in WASO and WASOTO6 were noted on Night 1, but these values did not reach statistical significance. Numerically greater improvements in sleep duration, as measured by TST and SE, were observed for

**Table 2. Summary of Change in Latency to Persistent Sleep.**

| Parameter | Placebo | Tasimelteon 20 mg | Tasimelteon 50 mg |
|---|---|---|---|
| N | 104 | 108 | 109 |
| **Baseline** | | | |
| Mean (SD) | 78.2 (45.8) | 78.8 (38.5) | 76.417 (39.7) |
| Median | 67.5 | 70.5 | 67.0 |
| Min, Max | 30.88, 393.5 | 30.3, 214.3 | 30.0, 204.0 |
| **Change from Baseline to the Average of Night 1 and Night 8** | | | |
| Estimate Mean Change (SEM) | −28.3 (3.0) | −44.9 (2.9) | −46.4 (2.9) |
| 95% CI | (−34.2, −22.4) | (−50.8, −39.2) | (−52.2, −40.6) |
| Active-Placebo Difference | | −16.7 | −18.091 |
| 95% CI | | (−24.9, −8.4) | (−26.3, −9.9) |
| *p*-value (vs. placebo) | | 0.000086 | 0.0000208 |
| **Change from Baseline to the Average of Night 22 and Night 29** | | | |
| Estimate Mean Change (SEM) | −33.9 (3.3) | −49.4 (3.3) | −45.1 (3.3) |
| 95% Confidence Interval | (−40.4, −27.3) | (−55.9, −42.9) | (−51.5, −38.6) |
| Active-Placebo Difference | | −15.5 | −11.2 |
| 95% Confidence Interval | | (−24.7, −6.3) | (−20.3, −2.0) |
| *p*-value (vs. placebo) | | <0.001 | 0.016 |

p-value (two-sided) from Pairwise Chi-square comparison of each active group (Tasimelteon 20 mg or 50 mg) to placebo based on an MMRM model including no intercept term, the effect of treatment over time, the pooled center effect and an interaction term between the adjusted baseline score and time was included as a covariate.

Abbreviations: SD = Standard Deviation; SEM = Standard Error of the Mean; CI = confidence interval

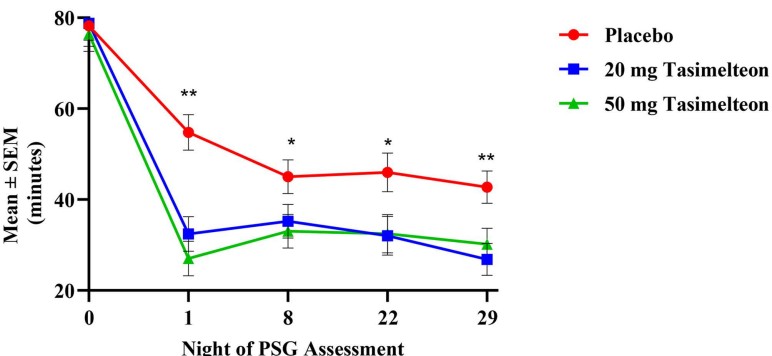

**Latency to Persistent Sleep**

Legend:
- ● Placebo
- ■ 20 mg Tasimelteon
- ▲ 50 mg Tasimelteon

X-axis: Night of PSG Assessment (0, 1, 8, 22, 29)
Y-axis: Mean ± SEM (minutes)

**Fig 2. Individual In-Clinic Night Assessment of Latency to Persistent Sleep.** Data points represent the mean ± SEM of polysomnography (PSG) sleep analysis for each treatment group at Baseline (N0) and during treatment administration on Nights 1, 8, 22 and 29. Data was analyzed by Mixed-Model Repeated Measures (MMRM) model. Asterisk (*) represent the significant differences between the tasimelteon 20 mg group and placebo group. *p value < 0.05; ** p value < 0.01 (2 tailed).

**Table 3. Summary of Effect of Tasimelteon on Latency to Persistent Sleep and Subjective Sleep Latency by Evaluation Night.**

| Parameter: | Placebo | Tasimelteon 20 mg | Tasimelteon 50 mg |
|---|---|---|---|
| N | 104 | 108 | 109 |
| **Change from Baseline to Night 1/Day 2** | | | |
| LPS[a] | −23.4 (3.9) ** | −46.4 (3.8) ** | −49.4 (3.8) ** |
| sSL[b] | −16.0 (59.8) ** | −31.9 (36.4) ** | −25.6 (33.0) * |
| **Change from Baseline to Night 8/Day 9** | | | |
| LPS | −33.1 (3.7) | −43.6 (3.7) * | −43.4 (3.7) |
| sSL | −22.2 (66.3) | −32.2 (39.6) | −22.6 (62.7) |
| **Change from Baseline to Night 22/Day 23** | | | |
| LPS | −32.2 (4.3) * | −46.3 (4.3) * | −43.9 (4.2) |
| sSL | −23.1 (41.9) | −32.3 (39.1) * | −25.8 (37.2) |
| **Change from Baseline to Night 29/Day 30** | | | |
| LPS | −35.5 (3.6) ** | −51.9 (3.5) ** | −46.2 (3.5) * |
| sSL | −28.3 (40.6) | −35.3 (37.8) | −34.2 (33.6) |

[a]p-value (2-sided) from Pairwise Chi-square comparison of each active group (20 mg or 50 mg) to placebo based on an MMRM model including no intercept term, the effect of treatment over time, the pooled center effect and an interaction term between the adjusted baseline score and time was included as a covariate.

[b]p-value (two-sided) for the pairwise comparison of each active group (20 mg or 50 mg) to placebo based on an ANCOVA model that includes main effects for treatment, (pooled) site and the adjusted baseline assessment as a covariate. The interaction between treatment and baseline was found to be significant at the 0.10 level and was thus left in the model. The pairwise p-values for the model without cut-offs for baseline values must thus be interpreted with caution.

*p-value < 0.05; ** p-value < 0.01 (2-tailed). Abbreviations: LPS = latency to persistent sleep; sSL = subjective sleep latency

patients receiving either 20 mg or 50 mg tasimelteon when compared with patients receiving placebo at all time points, and TST was significantly different on Night 1 (Table 4). On the average of Nights 1 and 8 of treatment, mean TST improved by 51 min in the 20 mg tasimelteon treatment group, 52 min in the 50 mg tasimelteon treatment group, and 40 min in the placebo group. On the average of Nights 22 and 29 of treatment, mean TST improved by 60 min in the 20 mg tasimelteon

**Table 4. Summary of Effect on Sleep Maintenance and Duration by Evaluation Night.**

| Endpoint | Evidence of Benefit Time-frame | Tasimelteon 20 mg (Change from Baseline) | Difference from Placebo |
|---|---|---|---|
| WASOTO6 | Night 1 | 25 min | 9 min |
| | Night 8 | 17 min | 6 min |
| | Night 22 | 25 min | 4 min |
| | Night 19 | 25 min | 3 min |
| TST | Night 1 | 57 min | 21 min* |
| | Night 8 | 46 min | 2 min |
| | Night 22 | 60 min | 12 min |
| | Night 29 | 60 min | 12 min |
| SE | Night 1 | 12% | 5%* |
| | Night 8 | 10% | 1% |
| | Night 22 | 13% | 3% |
| | Night 29 | 13% | 3% |

p-value (2-sided) from Pairwise Chi-square comparison of each active group (20 mg or 50 mg) to placebo based on an MMRM model including no intercept term, the effect of treatment over time, the pooled center effect and an interaction term between the adjusted baseline score and time was included score by time as a covariate.

*Denotes p-value for comparison vs. placebo<0.01.

WASOTO6 = Wake After Sleep Onset through Hour 6; Sleep efficiency (SE) was defined as the total sleep time (TST), determined by polysomnography, as a percentage of the full night.

treatment group, 49 min in the 50 mg tasimelteon treatment group, and 47 min in the placebo group. On the average of Nights 1 and 8 of treatment, mean SE improved by 11% in the 20 mg tasimelteon treatment group, 11% in the 50 mg tasimelteon treatment group, and 8% in the placebo group. On the average of Nights 22 and 29 of treatment, mean SE improved by 13% in the 20 mg tasimelteon treatment group, 10% in the 50 mg tasimelteon treatment group, and 10% in the placebo group. These improvements did not reach statistical significance.

### No Rebound, withdrawal or next-day residual effects

Rapid discontinuation of tasimelteon after 5-weeks of use did not appear to result in a worsening of any baseline insomnia symptoms (sleep onset, maintenance, or duration). During the placebo wash-out, insomnia patients receiving 20 mg or 50 mg tasimelteon did not continue to show the magnitude of improvement in sleep parameters observed during the double-blind evaluation interval, yet the values were still better compared to baseline values (S1 Table). Patients with insomnia randomized to placebo continued to improve in sleep parameters during the placebo wash-out interval when compared to their double-blind evaluation values. Moreover, abrupt discontinuation of 20 mg tasimelteon or 50 mg tasimelteon after 5 weeks of treatment did not appear to cause the types of subjective withdrawal symptoms experienced with benzodiazepines in pharmacologically dependent subjects (S2 Table).

No notable differences in next day residual effects, assessed as change from baseline to Days 2, 9, 23, and 30 by DSST and VAS, were observed the morning after treatment for insomnia patients receiving either 20 mg tasimelteon or 50 mg tasimelteon when compared with patients receiving placebo (S2 Table).

### Safety of tasimelteon in patients with primary sleep-onset insomnia

In this study, VEC-162 was safe and well-tolerated at the doses studied. The most frequent TEAEs (>3% of subjects in any treatment group) included headache, nasopharyngitis, blood CPK increased, and urinary tract infection.

Two treatment-emergent SAEs were reported for 2 patients in this study; 1 patient was in the 20 mg tasimelteon treatment group and 1 patient in the placebo group. Ten TEAEs were reported for 6 subjects that resulted in discontinuation

from the study; 2 patients from each treatment group (tasimelteon 20 mg, tasimelteon 50 mg and placebo group) (S3 Table).

The incidence of clinically notable laboratory abnormalities was low overall and similar between the placebo and the tasimelteon treatment groups. Low neutrophils, high eosinophils, and high ALT values outside the extended normal range were more common in the tasimelteon treatment groups compared with the placebo group.

For each vital sign parameter, the incidence of clinically notable abnormal values was low for all treatment groups, and similar in the placebo and the tasimelteon treatment groups and between baseline and post-treatment. No clinically meaningful differences between treatment groups in the occurrence of ECG abnormalities were observed.

## Discussion

Insomnia is the most common clinical sleep complaint and has a heterogenous symptom profile which encompasses difficulty in initiating sleep, maintaining sleep, and/or experiencing non-restorative poor-quality sleep leading to next day impacts on the individual and societal level [2]. While the causes of sleep-onset insomnia are not fully understood, it is probable that in many cases a primary or contributing factor is misalignment of the sleep-wake cycle with respect to desired bedtime. An optimal therapeutic intervention for this population of patients with insomnia would thus exhibit both a circadian regulatory effect and a sleep promoting effect, in which the circadian regulatory effect would work to rapidly phase shift the patient's endogenous circadian pacemaker and thereby realign the sleep-wake propensity rhythm with the desired bedtime [10]. Although traditional hypnotics have been shown to promote sleep, these agents do not affect circadian adaptation [10,11].

Tasimelteon is a circadian regulator that can reset the circadian pacemaker in the suprachiasmatic nucleus (SCN) [20]. The SCN of the hypothalamus controls various cyclical physiological functions of th body including the melatonin and cortisol circadian rhythms, the cyclical variation of core body temperature, and the sleep wake cycle [21]. Tasimelteon is selective for the MT 1and MT2 receptors, and it exhibits full agonist activity at these receptors [14]. In the clinical development of tasimelteon for Non-24 Sleep Wake Disorder and Jet-lag disorder (JLD), tasimelteon has been shown to phase advance and entrain the circadian pacemaker, as demonstrated by the circadian rhythmic release of melatonin and cortisol, as well as increase the amount of night-time sleep, decrease in the duration of daytime sleep and improvements in daytime functioning [18,19,22].

In this study, tasimelteon decreased the latency to persistent sleep for patients with chronic primary insomnia beginning on the first night of treatment, and this effect persisted for the 4-week duration of the study. The pre-specified primary outcomes of this study were met as evidenced by the immediate and persistent improvements in LPS for patients receiving either 20 mg or 50 mg tasimelteon when compared with patients receiving placebo. The currently approved dose of tasimelteon (20 mg) was able to reduce the amount of time it took to fall asleep for patients with chronic insomnia. Moreover, this clinically significant improvement was maintained with a reduction in sleep initiation of 52 minutes at 4 weeks.

A strong placebo response was observed for LPS in this study, higher than in many other studies of treatments for patients with insomnia [23]. This may potentially be explained by the regression toward the mean, since an extended sleep latency was a requirement for inclusion within the study. However, the response to tasimelteon, at both doses investigated in this study, was still significantly larger than the strong placebo response, highlighting the meaningful and clinically important impact of tasimelteon in improving latency to persistent sleep for patients with chronic sleep onset insomnia.

In this population of chronic primary insomnia patients, the benefits of tasimelteon appear to lie in improving initiation of sleep, *vs.* improving maintenance of sleep, thus the use of tasimelteon to treat insomnia characterized by difficulties with sleep initiation would be optimal. While cognitive behavioral therapy for insomnia (CBT-I) is currently first-line therapy, pharmacologic therapy is frequently incorporated in treating chronic and acute insomnia [24]. However, it is important to note that all prescription drugs approved for insomnia have adverse effects including, but not limited to, addiction, withdrawal, tolerance, and next-day adverse residual effects, including driving impairment and sleep-driving [25–27].

Non-benzodiazepine agents (e.g., Ambien®; Sonata®) are the most popular choice for the prescription treatment of insomnia but are only approved for the short-term treatment of insomnia due to the risks associated with long-term use [28]–[29][30]. Dual orexin receptor antagonists (DORAs) [31] (e.g. Belsomra®, Dayvigo®, Quviviq®) have reduced dependency or withdrawal symptoms, unlike other prescription sleep aids [32,33]. However, next-day residual effects are mixed; impairments on next-day driving have been reported [34], and sleep-driving represents a potentially serious side effect of daily usage.

Ramelteon (Rozerem®), like tasimelteon, is a melatonin-receptor agonist that has been approved for treatment of patients with sleep-onset insomnia by the FDA [35]. However, ramelteon shows just modest efficacy in this population, with large interindividual variability [36]. This is likely due to poor bioavailability (1.8% for ramelteon [37]), which limits its utility. In contrast, tasimelteon has 38.3% bioavailability [38]. Ramelteon, like melatonin, shows higher MT1 binding affinity, which may be a contributing factor to the soporific effects reported for these drugs [37]. In addition, comparable functionality to prolonged-release melatonin is provided by the ramelteon active metabolite MT-II, resulting in similar exposure duration [39]. Tasimelteon, in contrast, preferentially binds the MT2 receptor, believed to be most essential for phase-shifting the circadian clock [20].

Melatonin is available in the U.S. as an over the counter (OTC) food supplement in many doses and forms [40], but the lack of standardized manufacturing (26% of supplements also contained serotonin [41]), coupled by its inherent poor bioavailability (3% for melatonin [42] *vs* 38.3% for tasimelteon) make its effects unpredictable [36]. Nonetheless it remains a popular, and rising, choice among insomnia patients [43].

At present, all prescription drugs approved for insomnia have adverse effects including, among other things, addiction, withdrawal, tolerance, and next-day residual effects including driving impairment and sleep-driving [25–27,44]. Tasimelteon, by contrast, is a safe, effective and tolerable drug with no evidence of addiction, rebound insomnia, or next-day residual effects, including no next-morning residual effects on simulated driving performance [45].

The Diagnostic and Statistical Manual of Mental Disorders, Fifth Edition [2] contemplates an insomnia diagnosis if symptoms occur at least 3 nights per week, meaning many patients diagnosed with insomnia do not experience symptoms, nor need pharmacological treatment, every night of the week. Estivill *et al.*, reported that, from a survey of 1515 insomnia patients, 60% of patients did not experience symptoms every night. Moreover, only 56% took medication for insomnia symptoms every night of the week [46]. In addition, it has been shown that extended daily use of insomnia drugs may be linked to a higher risk of mortality [47] and impacts sleep stages with corresponding effects on sleep quality. Daily use of insomnia drugs can also lead to the development of drug tolerance, leading to the need for higher doses of the drug over time to get the same clinical impacts [24]. This reality can result in addiction or withdrawal symptoms associated with the current FDA approved sleep drugs most frequently prescribed, leading to further development of rebound insomnia, anxiety, irritability, or strange dreams. Consequently, to lower the risk of developing tolerance or addiction, physicians are encouraged to prescribe these sleep aids for only a few nights a week [48]. Tasimelteon has been shown to be able to set and reset the circadian rhythm in patients with Non-24 [18]. Unlike Non-24, however, entrainment does not appear necessary for insomnia. Instead, a circadian regulator that works on the first night of treatment to set and/or reset the circadian rhythm when symptoms intermittently arise, in addition to exhibited soporific effects, may provide a personalized therapeutic fit for the subpopulation of insomnia patients with intermittent difficulties falling asleep due to circadian misalignment with their scheduled bedtime.

Furthermore, no evidence of tachyphylaxis has been reported for tasimelteon. Due to the rapid clearance of tasimelteon (half-life $= 1.3 \pm 0.4$ hours) [38], a steady state is never achieved, even if taken daily. Thus, the clinical impact of tasimelteon is likely not the result of drug accumulation. In combination with the highly significant first night clinical benefits presented above, it is reasonable to assume that intermittent usage, when needed, of tasimelteon is effective in treating sleep-onset insomnia. Because tasimelteon has been shown to improve sleep latency measurements from the first night of treatment and its effect is not the result of drug accumulation, consistent daily usage is not required for clinical impact.

Overall, the results of this double-blind placebo-controlled clinical trial demonstrate the efficacy of tasimelteon in the treatment of patients with chronic sleep onset insomnia, beginning with the first night of treatment. The results of the study strongly suggest that tasimelteon may be an effective therapeutic tool in the treatment of these individuals.

## Supporting information

**S1 Protocol. Insomnia Protocol. The finalized trial protocol (Version 2.0) dated August 20, 2007.**
(PDF)

**S2 Appendix. CONSORT Checklist.**
(PDF)

**S1 Table. Summary of Rebound and Withdrawal Effects.**
(PDF)

**S2 Table. Summary of Next-Day Residual Effects.**
(PDF)

**S3 Table. Summary of Treatment-Emergent Adverse Events.**
(PDF)

## Acknowledgments

We thank the participants of the VP-VEC-162–3104 for their commitment to helping us in find a new treatment for insomnia. We thank all principal investigators, and the site study staff involved in VP-VEC-162–3104. We thank our colleagues at Vanda Pharmaceuticals for their support in conducting the study.

## Author contributions

**Conceptualization:** Gunther Birznieks, Mihael H Polymeropoulos.

**Data curation:** Christos M Polymeropoulos, Mihael H Polymeropoulos.

**Formal analysis:** Changfu Xiao.

**Methodology:** Changfu Xiao, Gunther Birznieks, Mihael H Polymeropoulos.

**Project administration:** Gunther Birznieks, Mihael H Polymeropoulos.

**Software:** Changfu Xiao.

**Supervision:** Christos M Polymeropoulos, Gunther Birznieks.

**Validation:** Changfu Xiao.

**Visualization:** Naoise C Synnott, Changfu Xiao.

**Writing – original draft:** Naoise C Synnott.

**Writing – review & editing:** Naoise C Synnott, Christos M Polymeropoulos, Changfu Xiao, Gunther Birznieks, Mihael H Polymeropoulos.

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
