## [Decision Letter · Decision Letter 0]

2 Feb 2025

Dear Dr. Synnott,

Thank you for submitting your manuscript to PLOS ONE. After careful consideration, we feel that it has merit but does not fully meet PLOS ONE’s publication criteria as it currently stands. Therefore, we invite you to submit a revised version of the manuscript that addresses the points raised during the review process.

The reviewers have addressed some important aspects both methdologically as concerning the relevance of the results especially as the study has been conducted a long time ago and I would ask you to very carefully consider the reviewers' comments in your revision.

We look forward to receiving your revised manuscript.

Kind regards,

Sascha Köpke

Academic Editor

PLOS ONE

Journal Requirements:

2.   We note that the original protocol file you uploaded contains a confidentiality notice indicating that the protocol may not be shared publicly or be published. Please note, however, that the PLOS Editorial Policy requires that the original protocol be published alongside your manuscript in the event of acceptance. Please note that should your paper be accepted, all content including the protocol will be published under the Creative Commons Attribution (CC BY) 4.0 license, which means that it will be freely available online, and any third party is permitted to access, download, copy, distribute, and use these materials in any way, even commercially, with proper attribution.

Therefore, we ask that you please seek permission from the study sponsor or body imposing the restriction on sharing this document to publish this protocol under CC BY 4.0 if your work is accepted. We kindly ask that you upload a formal statement signed by an institutional representative clarifying whether you will be able to comply with this policy. Additionally, please upload a clean copy of the protocol with the confidentiality notice (and any copyrighted institutional logos or signatures) removed.

“Vanda Pharmaceuticals funded this work”

4. Please ensure that you refer to Figure 2 in your text as, if accepted, production will need this reference to link the reader to the figure.

5. Please upload a copy of Figure 3, to which you refer in your text on page 14. If the figure is no longer to be included as part of the submission please remove all reference to it within the text.

6.  Thank you for stating the following in the Competing Interests section:

“All authors are employees of Vanda Pharmaceuticals. MHP is the CEO of Vanda Pharmaceuticals.”

7. We note that you have indicated that there are restrictions to data sharing for this study. For studies involving human research participant data or other sensitive data, we encourage authors to share de-identified or anonymized data. However, when data cannot be publicly shared for ethical reasons, we allow authors to make their data sets available upon request. For information on unacceptable data access restrictions, please see http://journals.plos.org/plosone/s/data-availability#loc-unacceptable-data-access-restrictions.

8.  Please include your full ethics statement in the ‘Methods’ section of your manuscript file. In your statement, please include the full name of the IRB or ethics committee who approved or waived your study, as well as whether or not you obtained informed written or verbal consent. If consent was waived for your study, please include this information in your statement as well.

Reviewers' comments:

Reviewer's Responses to Questions

**Comments to the Author**

1. Is the manuscript technically sound, and do the data support the conclusions?

Reviewer #1: Yes

Reviewer #2: Yes

2. Has the statistical analysis been performed appropriately and rigorously?

Reviewer #1: No

Reviewer #2: No

3. Have the authors made all data underlying the findings in their manuscript fully available?

Reviewer #1: Yes

Reviewer #2: No

4. Is the manuscript presented in an intelligible fashion and written in standard English?

Reviewer #1: Yes

Reviewer #2: Yes

Reviewer #1: Page 2 Abstract- Method: The sentence ‘Entry criteria selected for enrollment of primary insomnia patients with difficulty falling asleep’ requires revision.

Page 5: The reference number for approval by a central Institutional Review Board is to be stated.

Page 4: Full name for abbreviation SCN is to be provided e.g.suprachiasmatic nucleus.

Page 4: Typo melatnoin. MT1 Mt2 melatonin agonist is to be written as MT1 MT2 melatonin receptors agonist.

Page 5: This sentence ‘prior to the enrollment of the first patient (first posted on October 23, 2007)’ is unclear and requires further explanation.

Page 7: The exclusion criteria is to be stated.

Page 7: The number of subjects in the blocks is to be stated. For centrally, more information is to be provided e.g. who performed it from Vanda Pharmaceuticals Inc

Page 8: The terms ‘Double-blind evaluation interval and Single-blind Placebo Wash-out Night’’ are to be standardized with Figure 1b. All the assessment tools/inventories/questionnaires are to be indicated and denoted in the Figure 1b/Figure1b footnote.

Page 10: For the phrase ‘Potential for withdrawal symptoms, residual effects, and rebound effects of these two doses were investigated using validated questionnaires.’, the name of the questionnaires are to be stated.

Page 10: For the sample size calculation, sample size calculator/software or formula was used is to be stated.

Page 11: The statistical analysis software (and its version and publisher name) and the accepted statistical significant level is to be stated.

Page 11: More description/information on the MMRM and ANCOVA in the statistical analysis section is to be provided e.g. model description, covariance structure, covariates details, effect size index, multiple testing corrections/post hoc comparisons where applicable.

Page 13: Below Table 1, there is another row of demographic. This is to be omitted. BMI min-max is to be placed below the median.

Table 2, 3 & 4: Statistical tests are to denoted in the table footnote.

The descriptive statistics/results (tables/supplementary tables) at various nights/days are to be provided prior to further data management.

Full name for OC is to be stated e.g. observed cases (OC).

Page 15 Figure 2 title: Typo 'asterisk *p value < 0.05; * p value < 0.01 (2 tailed).' ** is missing.

Supplementary Table 2: The notation for *p-value < 0.05 and **p-value < 0.01 (2-tailed) is to be removed, as no significant differences were observed. A footnote to be added to indicate that no statistical differences were noted.

Figure chart is to be clearly labelled Figure 2.

Ensure all figures/tables are cited in the text.

Not all references conform to the journal format.

Reviewer #2: The authors, all employees of the company making the product evaluated in this study, conducted a double blind placebo controlled trial with a placebo controlled run in phase among selected individuals with sleep latency disorders to evaluate the effects of tasimeteon (Heltioz) (at two different doses) to improve Latency to Persistent Sleep (LPS) primarily through polysomnography. Results showed a statistically significant improvement of approxmately 15-20 minutes in LPS for both doses averaged at days 1 and 8 (and also noted later)l vs. placebo. Effect size is modest but was also seen in patient reported sleep latency, THere were essentially no differences in outcomes for other physiologic measures of sleep. Authors stated they assessed patent reported Sleep associated quality of life and function but these more important outcomes were not provided and calls into concerns the selective reporting of findings. In the absence of clinically meaningful outcomes the data focused solely on LPS is of little clinical importance (esp given the lack of benefit for other sleep time measures). AUthors also note safety of the medication and in the discussion highlight the safety profile though the labeling in the FDA inserts notes adverse effects including liver function abnormalities, respiratory and urinary tract infections and difficulties maintaining alertness including performing functions (hardly fully safe). The authors also used a placebo controlled run in phase which in-part reduced their initially eligible by about50% to the final number randomized and has the known effect of then only enrolling and analyzng indivduals likely to adhere, with fewer adverse effects and greater benefits. The mean age of individyuals was in the 40s thus little is known on older adults. The authors note that not all data is available because some is patent identifiable. THis can be readily corrected so that others can have full access as required by PLOS. It is particularly concerning because the data in this paper is only for sleep time frames and not patient inportant measures such as function and sleep related quality (and studies were mostly done in sleep labs...not very applicable). Additionally, the study was completed in 2008 (17 years ago!!!). It is not clear to this reviewer why the results would be published at this late date and given the sponsorship, author employment, analyses plan, limited data reporting etc calls into concern selected outcome reporting and analyses. The intro and discussion are written as basic advertisments for the drug. AUthors do not mention the current price in the US ranges from about $5000-$20,000 per month for most patients. Readers should know all of this information to help decide if the product is a good value given the benefits limited to physiologic sleep latency of about 15-20 minutes in highly selected patients, no impropvement in other sleep time measures, no data on sleep related function or quality of life and stated harms in the FDA label. For some it might be worth it. THis article does not provide full transparency.

**Do you want your identity to be public for this peer review?** For information about this choice, including consent withdrawal, please see our Privacy Policy

Reviewer #1: No

Reviewer #2: No

---

## [Author Response · Author response to Decision Letter 1]

14 May 2025

REVIEWER #1

Page 2 Abstract- Method: The sentence ‘Entry criteria selected for enrollment of primary insomnia patients with difficulty falling asleep’ requires revision.

Vanda Response

Updates test: Entry criteria emphasized enrollment of primary insomnia patients with confirmed difficulty falling asleep.

Page 5: The reference number for approval by a central Institutional Review Board is to be stated.

Vanda Response

This has been added.

Page 4: Full name for abbreviation SCN is to be provided e.g.suprachiasmatic nucleus.

Vanda Response

This has been updated on page 4.

Page 4: Typo melatnoin. MT1 Mt2 melatonin agonist is to be written as MT1 MT2 melatonin receptors agonist.

Vanda Response

Typo has been fixed.

Page 5: This sentence ‘prior to the enrollment of the first patient (first posted on October 23, 2007)’ is unclear and requires further explanation.

Vanda Response

This has been updated to (first posted to ClinicalTrials.gov on October 23, 2007)

Page 7: The exclusion criteria is to be stated.

Vanda Response

These have been included under ‘Patients’ section.

Page 7: The number of subjects in the blocks is to be stated. For centrally, more information is to be provided e.g. who performed it from Vanda Pharmaceuticals Inc

Vanda Response

This section has been updated.

Page 8: The terms ‘Double-blind evaluation interval and Single-blind Placebo Wash-out Night’’ are to be standardized with Figure 1b. All the assessment tools/inventories/questionnaires are to be indicated and denoted in the Figure 1b/Figure1b footnote.

Vanda Response

Footnote has been updated.

Page 10: For the phrase ‘Potential for withdrawal symptoms, residual effects, and rebound effects of these two doses were investigated using validated questionnaires.’, the name of the questionnaires are to be stated.

Vanda Response

These have been added.

Page 10: For the sample size calculation, sample size calculator/software or formula was used is to be stated.

Vanda Response

Software has been added.

Page 11: The statistical analysis software (and its version and publisher name) and the accepted statistical significant level is to be stated.

Vanda Response

Software has been added. A two sided alpha of 0.05 was already mentioned.

Page 11: More description/information on the MMRM and ANCOVA in the statistical analysis section is to be provided e.g. model description, covariance structure, covariates details, effect size index, multiple testing corrections/post hoc comparisons where applicable.

Vanda Response

This section has been expanded.

Page 13: Below Table 1, there is another row of demographic. This is to be omitted. BMI min-max is to be placed below the median.

Vanda Response

The table has been updated.

Table 2, 3 & 4: Statistical tests are to denoted in the table footnote.

Vanda Response

Footnotes have been updated.

The descriptive statistics/results (tables/supplementary tables) at various nights/days are to be provided prior to further data management.

Vanda Response

Please clarify what descriptive statistics are requested.

Full name for OC is to be stated e.g. observed cases (OC).

Vanda Response

This has been updated.

Page 15 Figure 2 title: Typo 'asterisk *p value < 0.05; * p value < 0.01 (2 tailed).' ** is missing.

Vanda Response

This has been updated.

Supplementary Table 2: The notation for *p-value < 0.05 and **p-value < 0.01 (2-tailed) is to be removed, as no significant differences were observed. A footnote to be added to indicate that no statistical differences were noted.

Vanda Response

This has been updated.

Figure chart is to be clearly labelled Figure 2.

Vanda Response

This has been updated.

Ensure all figures/tables are cited in the text.

Vanda Response

This has been updated.

Not all references conform to the journal format.

Vanda Response

These errors have been updated.

REVIEWER #2:

• The authors, all employees of the company making the product evaluated in this study, conducted a double blind placebo controlled trial with a placebo controlled run in phase among selected individuals with sleep latency disorders to evaluate the effects of tasimeteon (Heltioz) (at two different doses) to improve Latency to Persistent Sleep (LPS) primarily through polysomnography. Results showed a statistically significant improvement of approxmately 15-20 minutes in LPS for both doses averaged at days 1 and 8 (and also noted later)l vs. placebo. Effect size is modest but was also seen in patient reported sleep latency, THere were essentially no differences in outcomes for other physiologic measures of sleep. Authors stated they assessed patent reported Sleep associated quality of life and function but these more important outcomes were not provided and calls into concerns the selective reporting of findings. In the absence of clinically meaningful outcomes the data focused solely on LPS is of little clinical importance (esp given the lack of benefit for other sleep time measures).

Vanda response

Insomnia can be characterized by difficulties with sleep initiation, maintenance or early rising. As mentioned above, the population selected in this study was patients with primary insomnia associated with difficulties initiating sleep. This was done by two in-clinic PSGs which showed an LPS of >1 hour. As such, the main sleep compliant of this population is getting to sleep. Thus, our primary endpoint was looking at this. The study met the primary endpoint as defined before study initiation. As mentioned above, the latency to persistent sleep was significantly improved in the treatment groups across multiple time-points.

The changes in latency to persistent sleep are clinically meaningful and in line with the numbers seen by other sleep drugs on the market. Ambien, the most prescribed sleep aid in the US was approved by the FDA with similar changes in LPS (www.accessdata.fda.gov/drugsatfda_docs/nda/pre96/019908_S000_MOR.pdf )

Moreover, according to meta-analysis the impact of tasimelteon is more than twice the impact of melatonin (7 minutes) (1), one of the most widely used sleep aids, in reducing latency to persistent.

Additionally, a significant increase in total sleep time was also seen on night 1. Moreover, although other sleep measures did not achieve significance, they all showed numerical improvements over placebo.

On Nights 8, 22, and 29, no differences were seen between treatment and placebo for subjective sleep quality or debt. However, numerically greater improvements in subjective daytime function were observed for subjects receiving 20 mg tasimelteon when compared with subjects receiving placebo.

• AUthors also note safety of the medication and in the discussion highlight the safety profile though the labeling in the FDA inserts notes adverse effects including liver function abnormalities, respiratory and urinary tract infections and difficulties maintaining alertness including performing functions (hardly fully safe).

Vanda response

In this study the most frequent TEAEs (>3% of subjects in any treatment group) included headache, nasopharyngitis, blood CPK increased, and urinary tract infection. Tasimelteon has been on the US and EU market since 2014 with no updated safety issues, and the periodic safety update reports have all reported a positive benefit-risk balance throughout that time.

• The authors also used a placebo controlled run in phase which in-part reduced their initially eligible by about50% to the final number randomized and has the known effect of then only enrolling and analyzng indivduals likely to adhere, with fewer adverse effects and greater benefits.

Vanda response

The placebo run in was included in the study design to ensure a patient population with confirmed sleep onset insomnia was selected for.

• The mean age of individyuals was in the 40s thus little is known on older adults. The authors note that not all data is available because some is patent identifiable. THis can be readily corrected so that others can have full access as required by PLOS. It is particularly concerning because the data in this paper is only for sleep time frames and not patient inportant measures such as function and sleep related quality (and studies were mostly done in sleep labs...not very applicable).

Vanda response

Polysomnography is considered the gold standard method to measure sleep outcomes. Indeed, patient reported outcomes incredibly meaningful also. However, subjective measure of sleep latency also reported significant improvements in latency on Night 22, with numerical improvements vs placebo at all timepoints measured (Table 3).

• Additionally, the study was completed in 2008 (17 years ago!!!). It is not clear to this reviewer why the results would be published at this late date and given the sponsorship, author employment, analyses plan, limited data reporting etc calls into concern selected outcome reporting and analyses. The intro and discussion are written as basic advertisments for the drug. AUthors do not mention the current price in the US ranges from about $5000-$20,000 per month for most patients. Readers should know all of this information to help decide if the product is a good value given the benefits limited to physiologic sleep latency of about 15-20 minutes in highly selected patients, no impropvement in other sleep time measures, no data on sleep related function or quality of life and stated harms in the FDA label. For some it might be worth it. THis article does not provide full transparency.

Vanda response

As mentioned above tasimelteon has been approved in the US and Europe to treat other sleep disorder for over a decade. Based on the efficacy and safety data collected from the clinical development of Non-24 (2) and Smith Magenis Syndrome (3), tasimelteon has been shown to be a well-tolerated drug which can assist in sleep initiation and quality. These data are consistent with the data presented in this manuscript, and the mechanism of action as described in the manuscript introduction. Taken together with the clinical development program of tasimelteon and the consistency in LPS numerical changes to other drugs on the market, we believe this shows a meaningful result.

References

1. Ferracioli-Oda, E., Qawasmi, A., & Bloch, M. H. (2013). Meta-analysis: melatonin for the treatment of primary sleep disorders. PloS one, 8(5), e63773. https://doi.org/10.1371/journal.pone.0063773

2. Lockley SW, Dressman MA, Licamele L, et al. Tasimelteon for non-24-hour sleep-wake disorder in totally blind people (SET and RESET): two multicentre, randomised, double-masked, placebo-controlled phase 3 trials. Lancet. 2015;386(10005):1754-1764. doi:10.1016/S0140-6736(15)60031-9

3. Polymeropoulos CM, Brooks J, Czeisler EL, et al. Tasimelteon safely and effectively improves sleep in Smith-Magenis syndrome: a double-blind randomized trial followed by an open-label extension. Genet Med. 2021;23(12):2426-2432. doi:10.1038/s41436-021-01282-y

---

## [Decision Letter · Decision Letter 1]

23 Jul 2025

Dear Dr. Synnott,

Thank you for submitting your manuscript to PLOS ONE. After careful consideration, we feel that it has merit but does not fully meet PLOS ONE’s publication criteria as it currently stands. Therefore, we invite you to submit a revised version of the manuscript that addresses the points raised during the review process.

We look forward to receiving your revised manuscript.

Kind regards,

Sascha Köpke

Academic Editor

PLOS ONE

Journal Requirements:

Reviewers' comments:

Reviewer's Responses to Questions

**Comments to the Author**

Reviewer #1: (No Response)

2. Is the manuscript technically sound, and do the data support the conclusions?

Reviewer #1: Yes

3. Has the statistical analysis been performed appropriately and rigorously?

Reviewer #1: Yes

4. Have the authors made all data underlying the findings in their manuscript fully available?

Reviewer #1: Yes

5. Is the manuscript presented in an intelligible fashion and written in standard English?

Reviewer #1: Yes

Reviewer #1: Reference

14: JPharmSci Full name to be given,

44: JAMA, Full name to be given,

For Pairwise Chi-square contrast etc, the word contrast is to be replaced with another word.

The sentence 'The product of the adjusted baseline score by time as a covariate' requires revision e.g. an interaction term between the adjusted baseline score and time was included as a covariate

The sentence The treatment-by-baseline interaction’ could be rephrased as ‘interaction between treatment and baseline.’ This suggestion applies throughout the manuscript

**Do you want your identity to be public for this peer review?** For information about this choice, including consent withdrawal, please see our Privacy Policy

Reviewer #1: No

---

## [Author Response · Author response to Decision Letter 2]

31 Jul 2025

REVIEWER #1:

1. Reference

14: JPharmSci Full name to be given,

44: JAMA, Full name to be given,

Vanda Response

No. 14 has been updated to Journal of Pharmaceutical Sciences and no. 44 has been updated to Journal of the American Medical Association.

2. For Pairwise Chi-square contrast etc, the word contrast is to be replaced with another word.

Vanda Response

We have updated the text to: ‘p-value (two-sided) from Pairwise Chi-square comparison of each active group’.

3. The sentence 'The product of the adjusted baseline score by time as a covariate' requires revision e.g. an interaction term between the adjusted baseline score and time was included as a covariate

Vanda Response

We have updated the text to the suggested revision.

4. The sentence The treatment-by-baseline interaction’ could be rephrased as ‘interaction between treatment and baseline.’ This suggestion applies throughout the manuscript

Vanda Response

We have updated the text to the suggested revision.

---

## [Decision Letter · Decision Letter 2]

31 Aug 2025

Melatonin Agonist Tasimelteon (HETLIOZ®) Improves Sleep in Patients with Primary Insomnia: A multicenter, randomized, double-blind, placebo-controlled trial.

PONE-D-24-38898R2

Dear Dr. Synnott,

We’re pleased to inform you that your manuscript has been judged scientifically suitable for publication and will be formally accepted for publication once it meets all outstanding technical requirements.

Kind regards,

Sascha Köpke

Academic Editor

PLOS ONE

Additional Editor Comments (optional):

Reviewers' comments:

Reviewer's Responses to Questions

**Comments to the Author**

Reviewer #1: All comments have been addressed

2. Is the manuscript technically sound, and do the data support the conclusions?

Reviewer #1: (No Response)

3. Has the statistical analysis been performed appropriately and rigorously?

Reviewer #1: (No Response)

4. Have the authors made all data underlying the findings in their manuscript fully available?

Reviewer #1: (No Response)

5. Is the manuscript presented in an intelligible fashion and written in standard English?

Reviewer #1: (No Response)

Reviewer #1: (No Response)

**Do you want your identity to be public for this peer review?** For information about this choice, including consent withdrawal, please see our Privacy Policy

Reviewer #1: No

---

## [Editor Report · Acceptance letter]

PONE-D-24-38898R2

PLOS ONE

Dear Dr. Synnott,

I'm pleased to inform you that your manuscript has been deemed suitable for publication in PLOS ONE. Congratulations! Your manuscript is now being handed over to our production team.

Kind regards,

on behalf of

Professor Sascha Köpke

Academic Editor

PLOS ONE